# An Electroconductive, Thermosensitive, and Injectable Chitosan/Pluronic/Gold-Decorated Cellulose Nanofiber Hydrogel as an Efficient Carrier for Regeneration of Cardiac Tissue

**DOI:** 10.3390/ma15155122

**Published:** 2022-07-23

**Authors:** Hajar Tohidi, Nahid Maleki-Jirsaraei, Abdolreza Simchi, Fatemeh Mohandes, Zahra Emami, Lorenzo Fassina, Fabio Naro, Bice Conti, Federica Barbagallo

**Affiliations:** 1Department of Physics and Chemistry, Alzahra University, Vanak Village Street, Tehran 19938 93973, Iran; tohidi.biophysics@gmail.com; 2Department of Materials Science and Engineering, Sharif University of Technology, Azadi Avenue, Tehran 14588 89694, Iran; mohandes1120@gmail.com (F.M.); zahra.emami@sharif.edu (Z.E.); 3Institute for Nanoscience and Nanotechnology, Sharif University of Technology, Azadi Avenue, Tehran 14588 89694, Iran; 4Department of Electrical, Computer and Biomedical Engineering, University of Pavia, 27100 Pavia, Italy; lorenzo.fassina@unipv.it; 5Department of Anatomical, Histological, Forensic and Orthopedic Sciences, Sapienza University, 00185 Rome, Italy; fabio.naro@uniroma1.it; 6Department of Drug Sciences, University of Pavia, 27100 Pavia, Italy; bice.conti@unipv.it; 7Department of Experimental Medicine, Sapienza University, 00185 Rome, Italy; federica.barbagallo@uniroma1.it or; 8Faculty of Medicine and Surgery, Kore University of Enna, 94100 Enna, Italy

**Keywords:** smart hydrogel, gold-decorated cellulose nanofiber, nanocomposite, electroactive, heart diseases

## Abstract

Myocardial infarction is a major cause of death worldwide and remains a social and healthcare burden. Injectable hydrogels with the ability to locally deliver drugs or cells to the damaged area can revolutionize the treatment of heart diseases. Herein, we formulate a thermo-responsive and injectable hydrogel based on conjugated chitosan/poloxamers for cardiac repair. To tailor the mechanical properties and electrical signal transmission, gold nanoparticles (AuNPs) with an average diameter of 50 nm were physically bonded to oxidized bacterial nanocellulose fibers (OBC) and added to the thermosensitive hydrogel at the ratio of 1% *w/v*. The prepared hydrogels have a porous structure with open pore channels in the range of 50–200 µm. Shear rate sweep measurements demonstrate a reversible phase transition from sol to gel with increasing temperature and a gelation time of 5 min. The hydrogels show a shear-thinning behavior with a shear modulus ranging from 1 to 12 kPa dependent on gold concentration. Electrical conductivity studies reveal that the conductance of the polymer matrix is 6 × 10^−2^ S/m at 75 mM Au. In vitro cytocompatibility assays by H9C2 cells show high biocompatibility (cell viability of >90% after 72 h incubation) with good cell adhesion. In conclusion, the developed nanocomposite hydrogel has great potential for use as an injectable biomaterial for cardiac tissue regeneration.

## 1. Introduction

Cardiac tissue regeneration is difficult because of the low cell turnover rate of mammalian cardiomyocytes (around 0.3–1% annually) [1]. Therefore, heart implantation has so far been the clinical golden standard [2]. Nevertheless, the efficiency of transplantation is very low due to the limited number of donors, which brings out the need for innovative solutions to repair or replace injured cardiac tissues [2,3]. The development of tissue engineering or regenerative strategies could improve the rate of patient survival, which can be boosted by delivering cells, therapeutics, and engineered tissues to the point of care [3,4]. This approach is particularly important for preventing fibrous remodeling and scar formation [5,6]. Regenerative medicine for cardiac tissues relies on highly porous and three-dimensional (3D) scaffolds to mimic the extracellular matrix and, as a consequence, facilitate cell attachment, proliferation, differentiation, and vascularization for gas and nutrient transportation [7,8]. For efficient cell-based therapy, the cardiomyocyte maturation could be obtained via electrical, mechanical, or chemical stimuli [2,8]; to this regard, the unique features of injectable hydrogels for delivering cells and therapeutics have attracted vast interest in the field of cardiac repair after infarction [2,9]. The importance of this approach arises from the hydrogel’s ability to provide a matrix of support for cells, which overcomes the difficulties in cell retention [10]. Additionally, biomolecules, drugs, growth factors, and DNA plasmids can be encapsulated into the hydrogel matrix and be delivered with controlled release kinetics to promote endogenous cell recruitment without detrimental systemic effects [11,12,13].

In particular, injectable hydrogels can be prepared from a vast range of natural or synthetic biomaterials. Collagen, hyaluronic acid, fibrin, alginate, agarose, chitosan, keratin, matrigel, and decellularized ECM (extracellular matrix) are among the natural biopolymers that have been widely used to support cell activities [2]. Dai et al. [14] employed a collagen hydrogel to treat myocardial infarction in rats and demonstrated the benefits of the hydrogel for cardiac repair in vivo. Ifkovits et al. [15] and So et al. [16] modified hyaluronic acid with tetramethylethylenediamine/ammonium persulfate and PEG-SH_2_, respectively, and used them as a promising scaffold for cardiac tissue repair. They have shown that hydrogels can provide a significant improvement in stroke volume and ejection fraction along with reduced size and increased wall thickness of the infarcted area. The non-thrombogenic properties of calcium-crosslinked alginate hydrogels have also been employed for cardiac tissue engineering in small (rat) and large (porcine) animal models [17]. Kofidis et al. [18] examined embryonic stem cell-laden matrigel after injection into an infarcted myocardium and observed the effectiveness of the procedure to fix the heart’s functionality through enhanced vascularization and reduced cell death. Decellularized cardiac tissue hydrogels have also been prepared and injected into rats [19]. In vivo assays have affirmed their hemocompatibility without noticing embolization or ischemia.

Despite the biocompatibility and bioactivity of natural polymers, they mostly suffer from low mechanical strength, possible immune response, batch-to-batch variations, and difficulties in obtaining structural modifications [20]. Therefore, many synthetic polymers, which have been developed and used for tissue engineering and regenerative medicine, may potentially be used for cardiac repair. The importance of synthetic polymers arises from their strong mechanical durability and controllable features [21]; however, the synthetic polymers may show problems regarding their biocompatibility and biodegradability, and, as a consequence, they could cause an inflammatory response leading to scar formation [2,20]. Meanwhile, many attempts have been devoted to the fabrication of injectable hydrogels with temperature-responsive potential in order to form 3D scaffolds after injecting them into the defect site in situ [22,23]. In this regard, among various natural and synthetic alternatives, Pluronic^®^ F-127 (PF) and chitosan (CS) are widely used to fabricate injectable hydrogels because they are thermosensitive, biocompatible, and biodegradable [24,25,26].

In particular, PF, a tri-block copolymer with an average molecular weight of about 12,000 g/mol, is the most commonly used polymer for drug delivery system matrices. The lower critical solution temperature (LCST) of PF is around the physiological temperature that marks it as a proper candidate for preparing in situ forming hydrogels. The block copolymer forms micelles due to hydrophobic interaction [27]. Gratieri et al. [28] showed that gelling temperature for PF-based hydrogels varies between 22 °C and 32 °C depending on the concentration. The micelles are self-assembled by increasing temperature through the presence of physical interaction [25,29]. The combination of PF with a natural polymer, such as chitosan, would improve the micelle interactions by forming physical or chemical bonds with polymer chains. Park et al. [25] designed a thermosensitive hydrogel with conjugation of Pluronic^®^ F-127 on the chitosan surface. They have shown that the sol–gel transition temperature occurs at around 25 °C. They also reported the effect of chitosan on hydrophobic interactions as the main driving force for hydrogel formation. Jung et al. [30] prepared a temperature-responsive hydrogel by mixing hyaluronic acid and Pluronic F-127. They showed that the presence of hyaluronic acid between Pluronic micelles increases mechanical properties while decreasing gelling temperature.

Despite multiple benefits of PF and CS, weak mechanical properties and low chemical stability limit their applications after administration into the human body [31]. To overcome these issues, conjugated hydrogels have been developed to achieve enhanced mechanical properties and stability in physiological environments [23]. The mechanical strength and stability of conjugated hydrogels is commonly improved with the aid of chemical crosslinkers (i.e., glutaraldehyde [32], glucose [33], genipin [23], 1-ethyl-3-(3-dimethylaminopropyl) carbodiimide (EDC), and N-hydroxysuccinimide (NHS) [34]) through forming covalent bonds between polymer chains.

In addition, to meet the electrophysiological performance of cardiac tissues, many studies have employed nanoparticles of noble metals such as gold and silver to enhance the electrical conductivity of the hydrogels [35]. Li et al. [36] showed that Au nanoparticles (AuNPs) improve the cell adhesion of cardiac myocytes via localized stiffness. The presence of metallic nanoparticles not only enhances conductivity, but also reinforces stiffness [37], particularly of the elastic modulus [38]. However, the aggregation of ultrafine particles in the polymer matrix is likely to degrade mechanical properties [39]. Therefore, the conjugation of the nanoparticles on the surface of carboxylate- or amino-functionalized polymers [40], such as imidazolium-based polymers [41], porphyrin-based polymers [42], and bacterial cellulose nanofibers (BNC) [43], has been examined. Chen et al. [44] synthesized Au nanoparticles and deposited them on the surface of nanocellulose fibers. They demonstrated the superior catalytic properties of AuNPs. Meanwhile, further studies are required to explore the effect of gold-decorated nanocellulose fibers on the properties of thermo-responsive polymers.

This work presents the rheological behavior, gelation temperature, cytocompatibility, swelling, and biodegradation of electroconductive and thermosensitive hydrogel. The thermosensitive hydrogel was prepared by grafting carboxylated Pluronic (F-127) (CPF) to chitosan (CS) through EDC/NHS chemistry. EDC and NHS were used as crosslinking agents to activate the carboxyl groups for reaction with amino groups without any discoloration [45]. To induce electrical conductivity and to tailor the rheological and mechanical properties of the hydrogel, AuNPs were incorporated. Severe aggregation of the nanoparticles was avoided by in situ synthesis and stabilization on the surface of OBC nanofibers. Bacterial cellulose (BC) is a natural polymer with wide biomedical applications due to its biodegradability, biocompatibility, high tensile strength, and low thermal expansion coefficient [46,47]. The nanofibers were oxidized by 2,2,6,6-tetramethylpiperidine-1-oxyl (TEMPO) and mixed with a gold precursor to in situ formed and stabilized AuNPs. After synthesis, the effects of gold-decorated OBC on the rheological properties and electrical conductance of CS–CPF hydrogel were studied by dynamic rheology and different conductivity methods. The biocompatibility and degradability of the nanocomposite hydrogels were also investigated. Our results support the suitability of the developed nanocomposite hydrogel for cardiac repair.

## 2. Materials and Methods

### 2.1. Materials

Pluronic^®^ F-127 (PEO99–PPO65–PEO99) and medium molecular weight chitosan (190–310 kDa, 75–85% deacetylated) were purchased from Sigma Aldrich Chemicals (Merck KGaA, Darmstadt, Germany). An aqueous suspension of bacterial cellulose (1% *w*/*v*) was provided by Nano Novin Polymer Company, Sari, Iran. Succinic anhydride, 1-ethyl-3-(3-dimethylaminopropyl) carbodiimide (ECD), N-hydroxysuccinimide (NHS), 1,4-dioxane, triethylamine (TEA), 4-dimethylaminopyridine (DMAP), sodium hydroxide (NaOH), H_2_SO_4_ (95–98%), HCl (37%), NaCl (99.5%), KCl (99.5%), Na_2_HPO_4_ (99.0%), KH_2_PO_4_ (99.5%), ethanol (>99.9%), 2,2,6,6-tetramethylpiperidine-1-oxyl (TEMPO, 98%), sodium bromide (NaBr), three-ethanolamine (TEA), dioxane, and sodium hypochlorite solution (NaClO) were supplied by Sigma Aldrich Chemicals and used without purification. Phosphate buffer saline (PBS) was prepared by dissolving adequate amounts of the salts in DI water (18 MΩ.cm) and adjusting the solution to the desired pH (~7.4).

### 2.2. In Situ Synthesis of Gold Nanoparticles on Bacterial Cellulose Nanofibers

Ultrafine gold nanoparticles were formed and distributed in situ on OBC nanofibers according to the chemical reactions presented in Figure 1a. At first, oxidation of BC nanofibers was carried out according to the literature [45] by adding 1 g of BC nanofibers (1% *w*/*v*) into 100 mL of an aqueous solution containing 0.01 mmol of NaBr and 0.016 mmol of TEMPO. The NaClO solution (5 mM) was then added to the above suspension under stirring at pH = 10 for 60 min. At pH = 7, the mixture was washed twice with DI water and ethanol. Afterward, different amounts of HAuCl_4_ solution (20 and 75 mM) in the presence of the reducing agent (NaBH_4_, 6 mM) were added to the oxidized BC suspension under magnetic stirring at 4 °C. The reactions were left to complete for 2 h under continuous stirring. Functionalized OBC containing a gold concentration of 20 mM (Au20@OBC) and 75 mM (Au75@OBC) was utilized for further analyses. It is worth mentioning that at gold concentrations higher than 75 mM, cellular biocompatibility becomes inferior, while at concentrations less than 20 mM, conductivity is lost. Therefore, samples with these two concentrations were prepared and characterized.

### 2.3. Preparation of Injectable Nanocomposite Hydrogels

To prepare the thermosensitive CS/PF hydrogel, carboxylation of terminal hydroxyl groups of Pluronic block copolymers was performed (Figure 1b). The procedure was carried out at room temperature under continuous stirring in a dioxane solution (60 mL) containing succinic anhydride (250 mg), DMAP (260 mg), and TEA (300 mL) [16,17]. After 24 h, the solvent was removed by a rotary evaporator (Rotary evaporator, Heidolph Instruments GmbH & Co. KG, Schwabach, Germany), and the product was precipitated in cold diethyl ether and dried in a vacuum oven. The yield of the process was about 97%.

Chitosan was dissolved in acetic acid (0.1 M) to obtain a homogenous solution (1 g, 2.5%, *w*/*v*). This solution was mixed with CPF (10 g, 7 %, *w*/*v*) and chemical crosslinking was achieved via EDC/NHS chemistry (770 mg EDC/460 mg of NHS) at room temperature for 2 h (Figure 1c). Dialysis against DI water was carried out by a cellulose membrane (14 kDa, Sigma, St. Louis, MO, USA). Finally, lyophilization was performed at −80 °C for 24 h (Alpha 2,4, Martin-Christ Freeze Drying Systems GmbH, Osterode, Germany).

### 2.4. Material Characterizations

Microscopy: Transmission electron microscopy (TEM, JEOL, JEM-1010, Tokyo, Japan) and a field-emission scanning electron microscope (FE-SEM, TESCAN MIRA 3, Kohoutovice, Czech Republic) were employed to study the structure of prepared hydrogels. Dimensional features, including the size of AuNPs aternd diameter of pores, were analyzed by NIH ImageJ software. On average, 50 independent images from different locations were examined.

Spectroscopy: IR spectroscopy was performed to determine the functional groups and absorption spectra of the synthesized materials. Fourier transform infrared (FTIR) spectra were obtained on a Nicolet Magna IR-550 spectrometer (Spectralab Scientific Inc., Markham, ON, Canada) using KBr pellets in the range of 400–4000 cm^−1^. The absorption spectra were recorded on a Perkin Elmer Lambda 35 UV/Vis spectrophotometer.

### 2.5. Rheological Analysis

Rheological studies were carried out on a Physica MCR 302 Rheometer (Anton Paar Ltd., Graz, Austria). A parallel and plate viscometer with a 25 mm diameter and 0.5 mm gap was used. The limit of the linear viscoelastic region was determined by an amplitude sweep at a constant angular frequency of 1 Hz at 37 °C. In this region, dynamic frequency sweep measurements were carried out in the frequency range of 1 to 100 Hz at a fixed strain of 1%. The gap adjustment was set at a normal force of 0.1 N. The thermogelling properties of the hydrogels were evaluated by determining the storage modulus (G′) and loss modulus (G″) during a temperature sweep from 15 to 42 °C at a constant heating rate of 0.03 °C/s and at a frequency of 1 Hz. To avoid dehydration during testing, silicon oil was used.

### 2.6. Electrical Conductance

The electrical conductivity of temperature-responsive hydrogels was measured by the Ionic electrical conductivity meter (Metrohm, Herisau, Switzerland) with two-point and four-point probe devices (Sanat Nama Javan Co., Esfahan, Iran). The distance between the electrodes and the thickness of the samples were 2 and 5 mm in two-point and four-point probe methods, respectively. By applying a varying DC potential, the current was measured and conductance (δ) was determined by Ohm’s law. The electrical spatial resistance (ρ) of hydrogel-containing human embryonic stem cell-derived cardiomyocytes (derived cardiomyocytes (HESC-CM)) was also determined by a Trans Epithelial Electrical Resistance (TEER) device.

### 2.7. Determination of the Swelling Ratio

To evaluate the swelling ratio, cylindrical-shaped hydrogels with a diameter of 10 mm and a height of ~0.2 mm were prepared. The specimens were incubated in PBS at 37 °C for various times of up to 30 h. The weight of the dried hydrogels and the swollen ones (after removing the excess liquid) was determined. The swelling ratio (S) was calculated by using the following equation [48]:S = [(W_w_ − W_d_)/W_d_] × 100(1)
where W_w_ and W_d_ are the mass of swollen and dried hydrogel, respectively.

### 2.8. In Vitro Degradation

To evaluate in vitro gel degradation, the samples (~0.2 mm height and 10 mm diameter) were soaked in PBS (pH = 7.4) at 37 °C for up to 11 weeks. The specimens were taken out from the incubators at several time intervals, rinsed with deionized water, dried (40 °C/10 min), and weighted. The in vitro degradation rate was determined from the weight difference before and after soaking in the medium versus time.

### 2.9. Cytocompatibility and Cell Adhesion

Sterilization of the hydrogels before cell incubation was carried out using ethanol (70%) and UV radiation for 60 min. Cell viability was evaluated by the 3-(4,5-dimethylthiazol-2-yl)-2,5-diphenyltetrazolium bromide (MTT) assay [49] on the hydrogel extracts. The incubation time continued for up to 7 days, and the culture medium (Dulbecco’s Modified Eagle’s Medium and fetal bovine serum to a final concentration of 10%) was used as the control. Rat cardiac cells (H9C2) with a density of 1 × 10^4^ per mL were employed. At the selected time interval (3 and 7 days), 10 μL of the reagent (0.5 mg/mL) was added to each well and incubated for 4 h. The absorbance at 570 nm was measured by a microplate reader (BioTek 800 TS Absorbance Reader, Santa Clara, CA, USA) after adding DMSO (100 µL).

Live and dead assays were performed by employing myocardial infarction cells isolated from rat neonates (103 cells/mL). After cell incubation for 7 days, staining was carried out by using acridine orange (AO), propidium iodide (PI), and 4′,6-Diamidino-2-phenylindoe (DAPI). Cell imaging was performed using a fluorescent microscope (Motic, Itasca, IL, USA) in green (live) and red (dead) channels.

## 3. Results

### 3.1. Characterizations of Thermosensitive and Electroconductive Hydrogels

Bacterial cellulose nanofibers were oxidized by 2,2,6,6-tetramethylpiperidine-1-oxyl (TEMPO) to produce carboxylic acid on their chain surface. Representative SEM images of the OBC nanofibers are shown in Figure 2a–c. A three-dimensional (3D) network of disordered nanofibers is visible. The diameter of the fibers is about 100 nm on average. The nanofibers were then mixed with a gold precursor to form AuNPs and stabilize them on the surface of the OBC. Figure 2d–i shows FE-SEM and EDX analysis of the gold-decorated nanofibers. The nanoparticles have spherical morphology with sizes in the range of 20 to 50 nm, as evidenced by TEM studies (Figure 2j–l).

EDX mapping reveals that AuNPs are homogeneously distributed on the surface of the nanofibers, and their amounts increase with increases in the concentration of gold precursor. Interestingly, at the highest AuNP concentrations, a fine and uniform distribution of nanoparticles on the cellulose fibers is achieved. Therefore, the utilized procedure avoids severe particle aggregation, which is vital for the targeted application (heart tissue engineering). More examinations of the prepared thermosensitive and electroconductive nanocomposite hydrogels have determined a 3D porous structure (Figure 3). The pore sizes range from 1 to 6 μm. It is proposed that the presence of gold-decorated nanofibers affects the hydrogel structure due to electrostatic interactions beside covalent bonds.

Figure 4A shows FT-IR spectra of OBC and Au@OBC samples. The results indicate that, after oxidation, a new peak at 1729 cm^−1^ has emerged that can be attributed to the C=O stretching vibration of carboxylic acid groups [50]. The peak located at around 3400 cm^−1^ is assigned to the stretching vibration of hydroxyl groups (OH). The stretching mode of –CH_2_ and C–O–C bonds appears at 2899 and 1060 cm^−1^, respectively. Microscopic studies have indicated that introducing the gold precursor with the reducing agent forms ultrafine AuNPs on the surface of nanofibers (Figure 2). The spectrum of Au@OBC determines that the intensity of the characteristic band for the C=O stretching vibration of carboxylic acid groups decreases, indicating the favorable deposition of gold nanoparticles on the cellulose nanofibers through electrostatic interaction. UV/Vis spectroscopy also reveals relatively broad surface plasmon resonance (SPR) absorption at 527 nm (Figure 4B). Compared to AuNPs synthesized in the absence of OBC nanofibers with a broad absorption peak at 542 nm, a slight blue shift is noticed that could be due to the sensitivity of the SPR band to the surface interactions [51].

The conjugation mechanism of CS to CPF is based on the reactions between carboxylate groups of CPF and amino groups of CS using EDC/NHS chemistry. EDC and NHS were used as crosslinking agents to activate the carboxyl groups for reaction with amine groups without any discoloration [45]. As mentioned before, because of the LCST behavior of Pluronic copolymers, the aggregation of Pluronic micelles obtained by hydrophobic interactions leads to the formation of the hydrogel at elevated temperatures [52]. Figure 5 shows a schematic presentation of pre-gel formation (before heating) by EDC/NHS chemistry and the resulting hydrogel (after heating) as an outcome of micelle self-assembly. As seen in Figure 4a, the absorption peaks at 1108 cm^−1^ and 1243 cm^−1^ are attributed to the C–O–C stretching of the aliphatic ether group and twisting vibration of –CH_2_ in PF, respectively. The presence of carboxyl stretching vibration at 1732 cm^−1^ in CPF spectra indicates the successful carboxylation of terminal hydroxyl groups by succinic anhydride [23].

### 3.2. Rheological Analysis

Figure 6a,b show the dynamic viscoelastic behavior of the thermosensitive hydrogels containing different amounts of AuNPs (20 and 75 mM) as a function of temperature. The values for complex viscosity (η*), which is the total resistance to flow, are also presented. In these experiments, the amount of OBC nanofibers in the hydrogel was kept constant at 1 wt%. Pluronic^®^ F-127 is a thermosensitive polymer with a gelling temperature around physiological conditions; therefore, the storage and loss moduli (G′ and G″) of the conjugated polymer composite change with temperature, giving an insight into the structural stability of the hydrogels and the sol–gel transition [53]. The results demonstrate that both moduli (G′ and G″) are relatively stable with slight or gradual changes in temperatures up to 20 °C.

The values of the moduli are low and below 100 Pa. However, a steep change appears at higher temperatures. The hydrogels start to break down rapidly at higher temperatures by improving elasticity (a very sharp increase of G′) at high concentrations of AuNPs. The crossing values of G′ and G″, which indicate the sol–gel transition (TG), occur at 21.4 °C and 15 °C for 20 and 75 mM AuNPs, respectively. This finding indicates that the higher Au concentration shifts the crossover point to a lower temperature by affecting the microgel structure. It seems that the presence of AuNPs accelerates the formation of micelle self-assembly. Meanwhile, increasing the concentration of AuNPs does not significantly influence the viscous contribution, i.e., the G″ value is marginally influenced by the gold nanoparticles. The gelatinization peak of the hydrogels (i.e., when the maximum values are achieved) is attained at around 37 °C (20 mM Au) and 30 °C (75 mM) with a great difference in the storage modulus. Therefore, hydrogel containing a higher amount of AuNPs is more likely crosslinked and more stable than the other. The results of complex viscosity also illustrate the high-temperature sensitivity of the hydrogels. A very sharp change in η* occurs at around 25 °C and 21 °C for the hydrogels containing 20 mM and 75 mM AuNPs, respectively. Complex viscosity also recorded higher values for the concentrated AuNPs, indicating less viscous flow behavior. The rheological parameters are summarized in Table 1.

Shear rate sweep measurements exhibit non-Newtonian and shear-thinning behavior (Figure 6c). The viscosity of the hydrogels decreases by increasing the shear rate by two to three orders of magnitude, making them suitable for injection and self-healing [54]. The viscosity values are also within the applicable range for extrusion through nozzles [55]. The effect of AuNPs is distinguishable at relatively low shear rates (<3 s^−1^). To explore this effect, the following relationship was utilized [56,57]:η = K · γ^(m−1)^(2)
where m and K are the shear-thinning index and consistency index (viscosity at γ = 1 s^−1^), respectively. The values of these parameters are presented in Table 1. The much lower values of the shear-thinning index, in comparison to Newtonian fluids (m = 1), demonstrate the crosslinked nature of the macromolecules. At 20 mM Au concentration, the m value is very low (0.03), exhibiting strong shear-thinning behavior. This observation indicates that the networks of Au@OBC nanofibers produced by physical interactions are quickly disrupted by the shear rate. Such behavior, which has also been reported in other hydrogels (for example, alginate/halloysite nanotubes [58]), may be attributed to (i) the breaking of physical crosslinks in the polymer network, (ii) interactions between fillers and filler/matrix interaction, and (iii) the alignment of polymer chains along the direction of flow [59]. However, at the higher gold concentration (75 mM), the shear-thinning index falls in the range of most thermosensitive chitosan-based hydrogels [60]. Therefore, the network of Au75@OBC nanofibers should be harder to break, revealing the role of fillers in the formation of polymer corona around the ultrafine particles that hinder flow [61]. It is worth mentioning that cellulose nanofibers could be oriented under shear forces; hence, the shear-thinning behavior of the hydrogels could be intensified with a decrease in the Newtonian-behavior region.

Frequency sweep experiments were also employed to evaluate the relationship between G′ and G″ and the frequency in the linear viscoelastic region at 37 °C (Figure 6d). To avoid structural rupture and investigate the stability and stiffness of prepared thermosensitive hydrogels, frequency sweep tests were performed. The hydrogels exhibit a solid-like behavior (G′ > G″) over the measured range of frequencies, demonstrating the thermosensitive character of the hydrogels. The storage modulus of the hydrogels is stable against frequency changes which reveals the formation of an entangled fibrous network [62]. Herein, it is important to mention that incubation of the hydrogels at 37 °C for 24 h before testing resulted in permanent chemical crosslinking; hence, no crossover point is seen. Despite the storage modulus, G″ gradually changes with frequency, which can provide some insight into hybrid crosslinking [63]. The addition of Au@OBC nanofibers significantly enhances the moduli compared to the CS–CPF hydrogel, revealing the impact of Au@OBC nanofibers on the strength of the conjugated polymer matrix. Meanwhile, the modulus for native myocardium is reported to be about 0.02 to 0.5 MPa [64] and the ratio of G′ to G″ is in the order of 10, which falls in the range of natural tissues reported in previous studies [65,66].

### 3.3. Electrical Conductance

The transfer of electrical signals in native cardiac tissue is essential for the proper function of the heart. The electrical potential difference across the membrane of cardiac muscle cells is −90 mV (intracellular relative to extracellular) with a conductivity of 0.03 to 0.6 S/m [67]. The results of conductivity measurements are shown in Table 2. The ionic conductivity of the hydrogels increases about twofold by increasing the gold concentration from 20 mM to 75 mM. The four-point probe assay has determined values of 2 × 10^−3^ S/m for 20 mM AuNPs and 6 × 10^−2^ S/m for 75 mM AuNPs. Although the electrical conductivity of the heart muscle tissue is between 16 × 10^−2^ and 50 × 10^−2^ S/m [68], studies have shown that conductivity between 10^−6^ and 10^−2^ S/m is sufficient for tissue engineering applications [69]. On the other hand, Trans Epithelial Electrical Resistance (TEER) measurements indicate a value of 0.086 S/m for the electrical spatial resistance of CS–CPF–Au75@OBC hydrogel containing human embryonic stem cell-derived cardiomyocytes (HESC-CM). It is worth mentioning that conductivity was measured within the first 3 h, in which the in vitro degradation had marginally started. The results indicate that, after a week, the amount of degradation was high (>50%), disintegrating the continuity of the scaffold and thus leading to a great loss of conductivity. Despite this, the conductance of samples containing AuNPs was higher than the corresponding sample without the incorporation of gold-decorated nanofibers. These findings indicate that AuNPs embedded within the hydrogel improve electrical communication between adjacent heart cells [70].

### 3.4. Swelling and Biodegradation of Hydrogels

Figure 7 shows the results of in vitro swelling and biodegradation of the nanocomposite hydrogel containing 75 mM Au after soaking in PBS (pH = 7.4, 37 °C) as a function of time. The porous hydrogel quickly absorbs water and reaches maximum swelling capacity of about 867 g/g at equilibrium after 3 h (Figure 7a). Swelling capacity is an intrinsic property of the hydrogel and indicates the penetration of the solvent into the void space between the hydrophilic polymer chains [71]. The gradual decreasing behavior of the swelling ratio at prolonged times can be attributed to biodegradation, as described below.

To evaluate in vitro degradation, the hydrogel was incubated in the culture medium for various times of up to 90 days (Figure 7b). The in vitro degradation response shows an upward trend with a rapid fragmentation in the first 10 days. The degradation of the hydrogel during those first 10 days is more than 70%. After 90 days, the in vitro degradation of the hydrogel is almost completed.

### 3.5. Cell Viability and Cardiac Cell Function

In vitro cytotoxicity of the composite hydrogel was evaluated by MTT assay using H9C2 heart cells. Figure 8A shows the high cell viability (>90% in comparison to untreated cells), and thus biocompatibility, of the hydrogel. The materials used for the synthesis of the hydrogels, including chitosan, Pluronic, and OBC nanofibers, are well known as biocompatible biopolymers that have been widely used for biomedical applications and that possess the ability to support cell growth [72,73,74]. Nontoxicity of 50 nm spherical gold nanoparticles has also been reported [75]. Figure 8B shows the results of acridine orange–propidium iodide (AO/PI) staining that discriminates viable cells (green staining) and dead cells (red staining). As shown in the figure, most of the cells show green staining with minimal amounts of dead cells, supporting the results of the MTT assay (Figure 8A).

## 4. Conclusions

Regenerative medicine for cardiac tissues relies on highly porous and three-dimensional (3D) scaffolds. To meet the electrophysiological performance of cardiac tissues, the addition of nanoparticles of noble metals is an effective strategy to enhance the electrical conductivity of hydrogels. Here, an injectable, thermosensitive, and electroconductive hydrogel was developed for cardiac tissue engineering. The hydrogel was formulated by chemical crosslinking of carboxylated Pluronic F-127 with chitosan through EDC/NHS chemistry. To improve mechanical durability and electroconductivity of the hydrogel, spherical AuNPs (50 nm) were synthesized and conjugated in situ on the surface of oxidized bacterial cellulose nanofibers with an average diameter of 100 nm. Microstructural studies revealed uniform distribution of Au-decorated OBC nanofibers in the CS–CPF matrix. Rheological studies demonstrated that with higher Au content, the gelation temperature was decreased. The hydrogel also exhibited more solid-like behavior with higher mechanical strength. The sol–gel transition occurred at 15–22 °C depending on the AuNP concentration. The swelling capacity of the developed hydrogel was 867 g/g after 3 h incubation in PBS, demonstrating its super porous structure. Moreover, in vitro degradation of the hydrogel in the culture medium displayed an upward trend for up to 10 days, in which about 70% of the mass was degraded. Electrical conductivity studies indicated the impact of AuNPs on conductance. A value of 6 × 10^−2^ S/m at 75 mM Au was attained by a four-point probe assay. MTT and live/dead assays using the H9C2 cardiomyocytes cell line affirmed the cytocompatibility of the hydrogel. In conclusion, the developed thermosensitive and electroconductive hydrogel has the potential to be employed as a platform for electroactive tissue repair.

## Figures and Tables

**Figure 1 materials-15-05122-f001:**
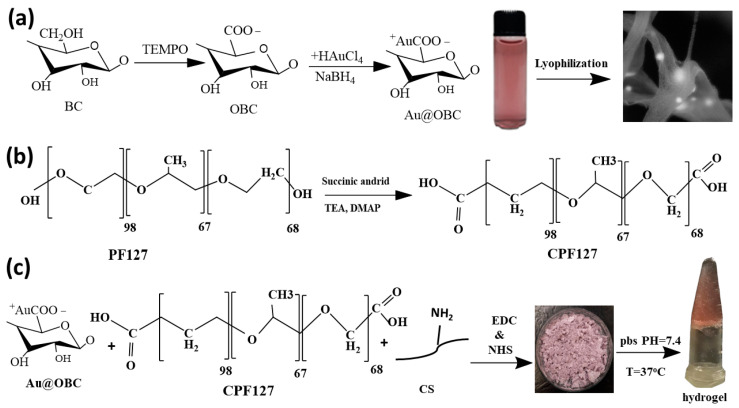
Schematic representation of the procedures employed for (**a**) in situ synthesis of AuNPs on bacterial cellulose nanofibers, (**b**) carboxylation of Pluronic F-127, and (**c**) production of electroconductive and thermosensitive nanocomposite hydrogels for cardiac tissue engineering.

**Figure 2 materials-15-05122-f002:**
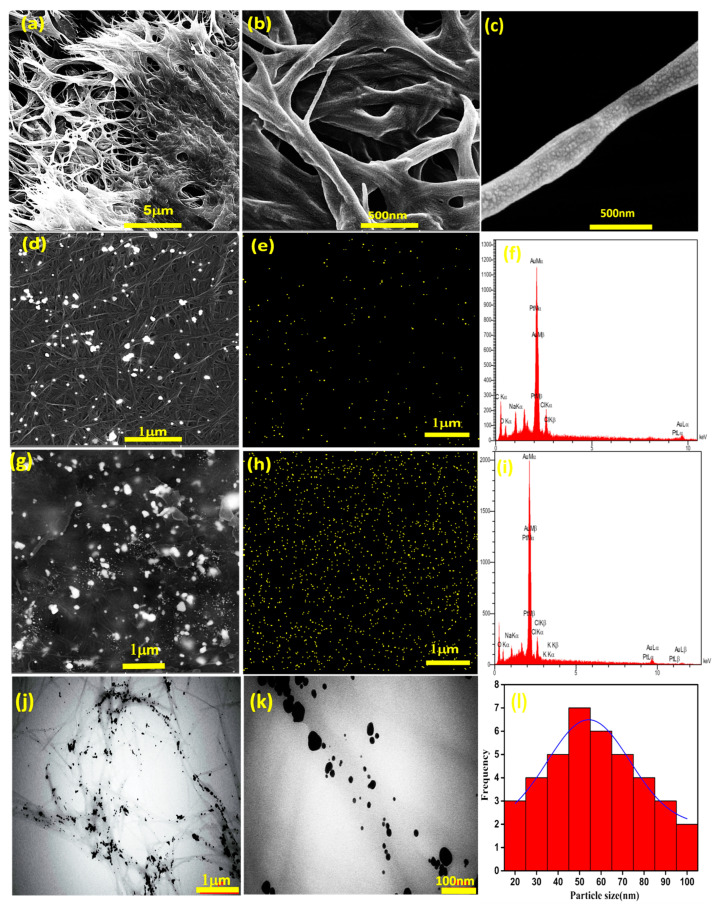
FE-SEM images of OBC nanofibers at different magnification showing (**a**) a porous 3D network, (**b**) an entangled fibrous structure, and (**c**) an individual fiber. (**d**) A FESEM image, (**e**) EDAX mapping, and (**f**) EDAX spectrum of Au20@OBC. (**g**) A FESEM image, (**h**) EDAX mapping, and (**i**) EDAX spectrum of Au75@OBC. (**j**,**k**) Representative TEM images of gold-decorated OBC nanofibers at different magnifications. (**l**) Particle size distribution of gold nanoparticles.

**Figure 3 materials-15-05122-f003:**
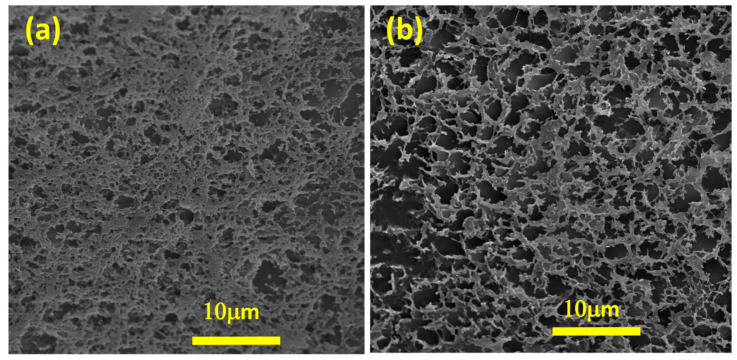
Representative FE-SEM images of (**a**) CS–CPF and (**b**) CS–CPF containing Au-decorated cellulose nanofibers.

**Figure 4 materials-15-05122-f004:**
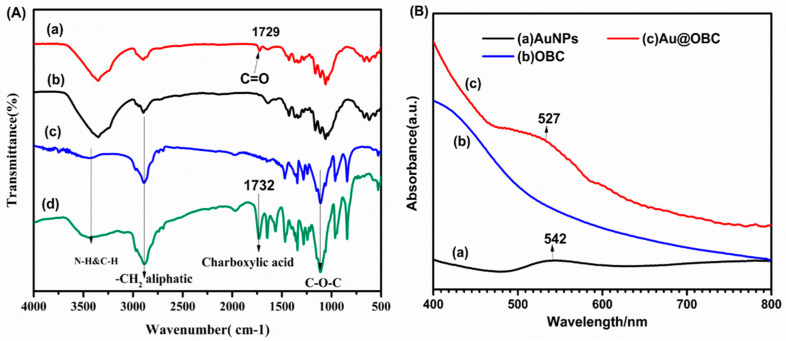
(**A**) FT-IR spectrum of (a) Au@OBC, (b) OBC, (c) BC, (d) PF. (**B**) UV-Vis absorption of aqueous suspensions of (a) unsupported AuNPs, (b) OBC, and (c) Au@OBC.

**Figure 5 materials-15-05122-f005:**
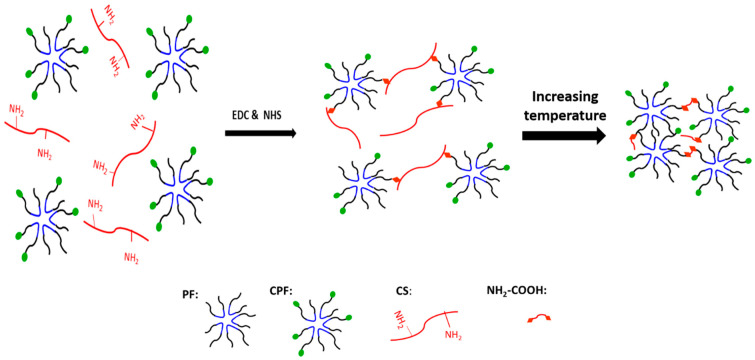
Schematic presentation of pre-gel formation (before heating), turning to a hydrogel after heating.

**Figure 6 materials-15-05122-f006:**
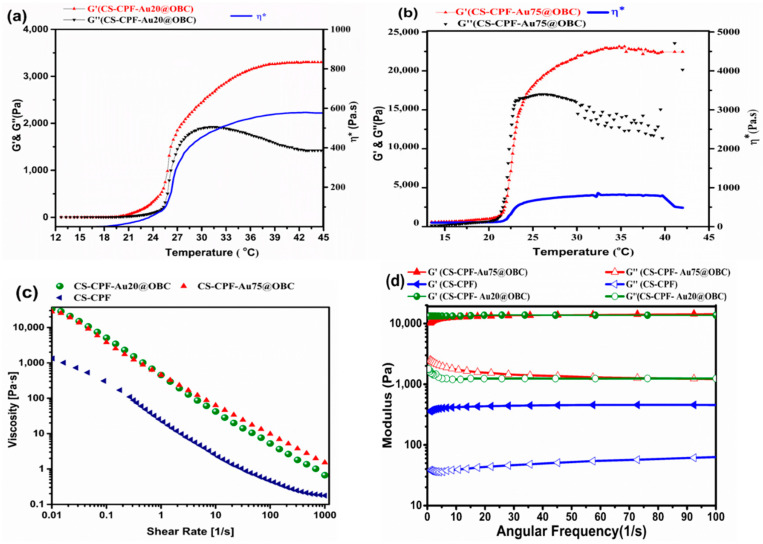
Results of temperature sweep tests for hydrogels containing (**a**) 20 mM and (**b**) 75 mM AuNP concentration. (**c**) Variation in viscosity versus the shear rate demonstrates the shear-thinning behavior of the hydrogels. (**d**) Results of frequency sweep tests.

**Figure 7 materials-15-05122-f007:**
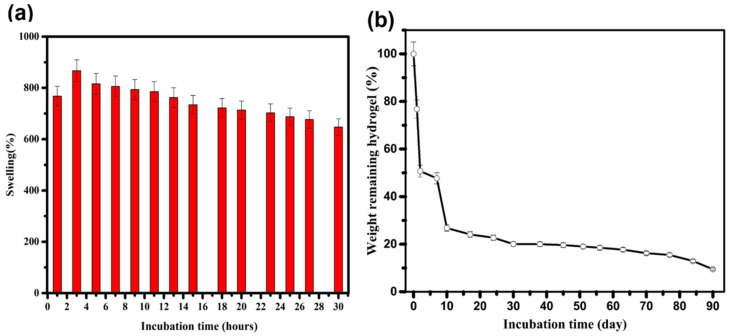
(**a**) Equilibrium swelling ratio (PBS, 37 °C, n = 4) and (**b**) in vitro degradation rate (culture medium, 37 °C, n = 3) of the thermosensitive and electroconductive hydrogel (75 mM AuNPs).

**Figure 8 materials-15-05122-f008:**
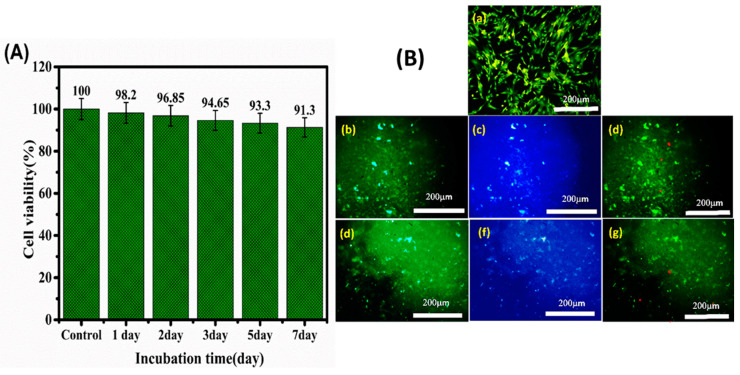
In vitro experiments related to cell survival and migration into the scaffolds. (**A**) Viability of H9C2 cells encapsulated within hydrogels for various incubation times. (**B**) Fluorescence microscopy images of the control test with H9C2 cells. (**a**) H9C2 cells encapsulated within hydrogels for various incubation times after (**b**–**d**) 3 days and (**e**–**g**) 7 days in vitro culturing, with AO/PI staining and DAPI/PI staining where green indicates live cells and the red stain indicates dead cells. Scale bars: 200 μm.

**Table 1 materials-15-05122-t001:** Rheological parameters of the examined thermosensitive and electroconductive hydrogels.

Au Concentration (mM)	T_G_ (°C)	$G^{\prime }\;(\mathrm{Pa})\;\mathrm{at}\;37\;{}^{\circ}C$	$G^{\prime \prime }\;(\mathrm{Pa})\;\mathrm{at}\;37\;{}^{\circ}C$	Shear-Thinning Index (m)	Consistency Index (Pa)
20	21.4	3300	1460	0.02	2.66
75	15	22,800	2460	0.21	1.67

**Table 2 materials-15-05122-t002:** The conductivity of thermosensitive and electroconductive hydrogels.

Method	CS–CPF-Au20@OBC	CS–CPF-Au75@OBC
Ionic conductivity	1.4 × 10^−4^ S/m	2.7 × 10^−3^ S/m
Four-point probe device	2 × 10^−3^ S/m	6 × 10^−2^ S/m
Two-point probe device	1 × 10^−3^ S/m	10^−2^ S/m
I–V	5 × 10^−4^ S/m	7 × 10^−3^ S/m
TEER device with HESC-CM	-	0.086 S/m

## Data Availability

Not applicable.

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
