# Peer review of "An Electroconductive, Thermosensitive, and Injectable Chitosan/Pluronic/Gold-Decorated Cellulose Nanofiber Hydrogel as an Efficient Carrier for Regeneration of Cardiac Tissue"

_materials, 2022, doi:10.3390/ma15155122_

Round 1
Reviewer 1 Report
Overall comments:
In the article authors developed a thermoresponsive injectable hydrogel using chitosan-poloxamer conjugate and AuNP decorated bacterial cellulose nanofibers. Authors have performed physicochemical and cell-based biocompatibility studies to evaluate the applicability of developed materials for cardiac tissue engineering application. This is an interesting piece of work, researchers working is respective fields might find it useful and informative.
However, I think extensive adjustments need to be made before reconsideration for publication.
Major comments:
1. Chemical structures in figure 1 need to be redrawn for correctness and uniformity.
2. Page 6 section 2.8
The term biodegradation involves degradation of a material either in presence of live biological cells or biomolecules (enzymes) originating from them. Tests were performed in absence of either of these components. Therefore the section should be appropriately renamed as “In-vitro degradation”.
3. Figure 2l legend is missing. Kindly correct.
4. Page 7 line 267
How the uniform particle size of AuNPs and, homogeneity and particle size distribution were measured. Data is missing.
5. Figure 6d: Information on loss modulus (G’’) of CS-CPF-Au75@OBC is missing. Kindly check.
6. Section 3.3: Whether the conductance measured in dry state or wet state of the hydrogel. What is the influence of degradation on conductance and the duration up to which conductance is maintained?
7. Data regarding day 14 of cell viability test (page 6, line 241) is missing from figure 8.
8. Section 3.5, kindly provide numerical information on %age cell viability at different time points. Also furnish higher quality images of cell tests, figure 8b.
9. Kindly check for correctness of reference 67. Check for correctness of all of the references.
Minor comments:
1. Page 4 section 2.3:
Details regarding name of the vendor and origin of the material needs to be complete and uniform throughout the manuscript. Kindly correct.
2. Page 13, line 432, kindly correct “cardiac tissues rely on” instead of “relays”. Check for typos throughout the manuscript.
3. Kindly check the references for repetition of reference 25, 29, 23 and 54.
Reviewer 2 Report
The work submitted for publication in the journal materials is of high interest due to its proposed application. In order to be published, it is recommended that the authors make some modifications to the paper.
The quality of the images is not the best and sometimes it is difficult to read the results on the graphs. It would be recommended to improve the quality of the graphics.
The personal form is used in some sections of the commentary on the results. In this type of text it is better to use the impersonal form. An example of this is found in line 272.
Figure 6a and 6b should have the x-axis with the same range and number of divisions because they are comparing equal results. Additionally the y-axis should be the same, but perhaps due to the large difference between the values of the two graphs probably if they are represented with the same scale figure 6a will be difficult to visualize.
In the international system the units of degree do not include the line. Also in line 349 "where" should be capitalized.
In Table 2, the superscripts of the conductivity values should be entered.
As mentioned in the electrical conductivity section, it would be interesting to mention whether the rheology values obtained in this work are in line with the results obtained by other authors for the applications proposed in this work.
Reviewer 3 Report
In the reviewed work authors obtained chitosan/poloxamer-based thermo-responsive and injectable hydrogel decorated by gold nanoparticles (AuNPs) physically linked to oxidized bacterial nanocellulose fibers (OBC) that could be considered for cardiac tissue regeneration. Porous structure with open-pore channels in the range of 50-200 µm has been formed, and shear rate sweep measurements confirm reversible phase transition from sol to gel. The obtained hydrogels display a shear-thinning behawior, as well as some electrical conductivity. In vitro cytocompatibility assays reveal high biocompatibility and good cell adhesion. The work is well-planned and methodologically sound, and the physico-chemical methods properly chosen, however, some issues need to addressed in more depth:
- Pluronic® F-127 – please name the selction criteria for this particular Poloxamer, as well as show its structure,
- As authors mention in the Introduction, „Despite multiple benefits of PF and CS, weak mechanical properties and low chemical stability limit their applications after administration into the human body”, please add some information on the mechanical properties of the obtained hydrogels,
- Table 1 – there are two Au concentrations showed that make any trend prediction difficult – please complete it and expand the discussion.
Reviewer 4 Report
This manuscript by Tohidi et al. describes the development and evaluation of polymeric hydrogels decorated with gold nanoparticles, as potential materials for treatment of heart disease. In my opinion, the work is described thoroughly and clearly, and merits publication.
I found only a couple of minor issues with the scientific or technical content; also, I did notice a number of typographical issues.
1) Subscripts and superscripts were not used in chemical formulae (e.g. H2SO4), units (e.g. density of 1×104 per mL) etc. I suggest the authors should carefully check their manuscript; otherwise these mistakes could propagate into the finished article.
2) The rheometer (MCR 302) was described as a 'disk-type cone and plate viscometer with 25 mm diameter and 0.5 mm gap '. The gap setting of 0.5mm strikes me as a very large value for a truncated cone. Do the authors mean a parallel plate geometry was used? Please clarify the text.
3) On lines 271-272, the pore sizes are quoted as 10 to 200 µm. But, that does not appear to be supported by the SEM images in Figure 3, which appear to suggest a range of about 1 to 6 µm. The authors should check, please.
4) Fig. 6: It would be useful to quote the frequency used for the temperature sweeps (6a and 6b) and the temperatures at which the shear rate and angular frequency sweeps were performed (6c and 6d).
Typographical issues (I have shown suggested corrections):
L30: '...shear modulus...'
L42: '...so far the clinical golden standard...'
L48: '...for cardiac tissues relies on highly...'
L159: I think the resistivity units should be megaohms/centimetre.
L182: '...crosslinking was achieved by the...'
L208: '...by determining the storage...'
L210: 'silicone oil'
L315-316: '...storage and loss moduli (G' and G")...'
L318: '...both moduli...'
L325: '...moduli...'
L402: '...and indicates the...'
